# Does AlphaFold2 model proteins' intracellular conformations? An experimental test using cross-linking mass spectrometry of endogenous ciliary proteins

Caitlyn L. McCafferty [1✉], Erin L. Pennington[1], Ophelia Papoulas [1], David W. Taylor [1✉] & Edward M. Marcotte [1✉]

A major goal in structural biology is to understand protein assemblies in their biologically relevant states. Here, we investigate whether AlphaFold2 structure predictions match native protein conformations. We chemically cross-linked proteins in situ within intact *Tetrahymena thermophila* cilia and native ciliary extracts, identifying 1,225 intramolecular cross-links within the 100 best-sampled proteins, providing a benchmark of distance restraints obeyed by proteins in their native assemblies. The corresponding structure predictions were highly concordant, positioning 86.2% of cross-linked residues within Cα-to-Cα distances of 30 Å, consistent with the cross-linker length. 43% of proteins showed no violations. Most inconsistencies occurred in low-confidence regions or between domains. Overall, AlphaFold2 predictions with lower predicted aligned error corresponded to more correct native structures. However, we observe cases where rigid body domains are oriented incorrectly, as for ciliary protein BBC118, suggesting that combining structure prediction with experimental information will better reveal biologically relevant conformations.

[1] Department of Molecular Biosciences, Center for Systems and Synthetic Biology, University of Texas, Austin, TX 78712, USA. ✉email: caitie.mccafferty@gmail.com; dtaylor@utexas.edu; marcotte@utexas.edu

The remarkable results of AlphaFold2 (AF2) in the 14th CASP competition[1] and the public release of code[2] has resulted in numerous applications for structural prediction[3–8]. While not the first attempt at proteome-wide structure prediction[9], AF2's success stems from its high accuracy at ab initio prediction[1]. Its broad applicability across protein families without requiring prior structural knowledge has already led to the discovery of at least 26 entirely new protein folds[10]. Global benchmarking and independent validation of such predicted structures will be necessary to inform reliable and nuanced interpretations of these structures.

Along with the AF2 method, an AlphaFold database was released that currently contains over 200 million protein structure predictions, including most of the Uniprot database[11]. Prior to AF2, it was estimated that the coverage of the human proteome by three-dimensional (3D) structures was about 48%; however, this fraction substantially increased to 76% with the inclusion of confident AF2 predictions. In addition, the dark proteome[12]—the set of proteins whose structures have not been observed experimentally and cannot be modeled with conventional homology modeling—shrank from 26 to 10%[13]. AF2 has similarly contributed to the increased coverage of disease-associated genes and mutations in the Clinvar database[14].

With the impressive boost in individual protein structure coverage, AF2 has also opened opportunities for structure prediction of protein–protein[15–17] and protein–peptide[18,19] interactions. For example, AlphaFold-multimer[15] offers a reasonably accurate prediction for many multi-protein complexes, as do several other tools that build on the original AF2 model[16]. Similarly, the addition of new protein structures has the potential to aid in the drug design of protein targets[20,21] and when combined with deep mutational scanning[22] can predict the effect of missense variants[8].

While these feats are impressive, caution is always merited in relying solely on computational predictions, and the degree of support for AF2 protein structures and regions with lower per-residue confidence scores (predicted Local Distance Difference Test, pLDDT <70) can be difficult to interpret. Moreover, it has been shown that AF2 predictions suffer for proteins that do not have available template sequences[21] and that AF2 is challenged by intrinsically disordered regions[23] and dynamics in general[24]. AF2 produces lower pLDDT scores in dynamic regions such as binding pockets[8], and it has been suggested that predictions of large multidomain proteins may not be suitable for drug studies[21].

There have been several studies demonstrating that combining AF2 structure predictions with experimental data can improve and aid in the interpretation of the results. Examples include combining predicted structures with cryo-EM data, crystallographic maps, or chemical cross-links[3,25,26]. When compared with NMR data, AF2 was better at predicting rigid loops, while NMR was superior in more dynamic regions[24]. Finally, AF2 has already proven useful for crystallographic phasing by molecular replacement[27]. Such studies suggest that combining computational predictions with experimental data can strongly increase confidence in and interpretability of the structure predictions.

We sought to independently assess AF2's confidence scores and ask if it correctly captured conformations of proteins in their cellular context. Because AF2 is trained on proteins in the Protein Data Bank[28], it could propagate biases present in that dataset, ranging from organismal biases to experimental techniques. However, AlphaFold2 also incorporates information from evolutionary coupling and amino acid conservation[1], which should, in principle, help to capture those structures most relevant to the predominant cellular roles of these proteins.

In our assessment of AF2, we therefore used in situ chemical cross-linking, performed on endogenous proteins directly within

their cellular contexts, as a method to provide 3D spatial information about proteins within their native conformations and assemblies. The use of XL/MS serves as independent experimental observations for pairwise distance restraints and can therefore be used in integrative modeling or as structural validation[29,30]. Cross-linking has the additional advantage of capturing distance restraints between residues in complex samples and within intact organelles—as we have demonstrated here–to complement other structural biology methods[31]. The use of this method to capture native contacts is especially fruitful when combined with methods such as cryo-EM. Previous studies combining these methods found that cryo-EM models have a 97% concordance with independently derived cross-links, assessed using the cross-linker BS3[32]. XL/MS can also be a useful tool in interpreting protein dynamics, where the recommended maximum length of 30 Å for cross-linkers DSS and BS3 is in agreement with empirically determined thresholds[33].

Our experimental dataset summarizes chemical cross-linking/mass spectrometry on intact cilia and native ciliary extracts isolated from Tetrahymena. Importantly, this organism has few experimentally determined protein structures. In fact, fewer than 60 experimentally determined Tetrahymena structures have been reported in the Protein Data Bank[28], 18 of which relate to telomerase[34]. The ciliary proteome is of particular interest because of its relevance to a wide range of human congenital disorders (ciliopathies)[35], and a better definition of ciliary protein structures is expected to offer insights into how specific alleles may lead to human disease. Importantly, ciliary proteins are highly conserved across eukaryotes, with many ciliary genes dating back to the last eukaryotic common ancestor[36]. Tetrahymena serves as a model organism for ciliary studies: It is easy to grow, roughly a thousand cilia decorate each cell, and large numbers of intact cilia can be prepared for biochemical analyses simply by treating cells with the nonlethal anesthetic dibucaine, which causes the cilia to detach from the cells[37].

In this study, we compare intramolecular distance restraints obtained from in situ chemical cross-linking and cross-links from enriched biochemical fractions of T. thermophila ciliary proteins to the AF2-predicted structures of the 100 most cross-linked proteins identified by mass spectrometry. In doing so, we hoped to address whether AF2, by incorporating co-evolutionary couplings[38,39], would have the power to detect biologically active structural conformations, especially for cases where multiple conformations or assembly states might occur. Our findings suggest that while there is a high concordance between our cross-links and AF2 structure predictions, we do observe violations between domains of multidomain proteins and those that undergo a dramatic conformational change.

## Results and discussion

We isolated intact cilia from T. thermophila[25,40] and cross-linked proteins directly within their native ciliary environments by using the membrane-permeable chemical cross-linker disuccinimidyl sulfoxide (DSSO), an analog of the cross-linker DSS that is also mass spectrometry cleavable. We supplemented these data with additional cross-links generated from native biochemical extracts of cilia after confirming the high agreement between these datasets (Supplementary Fig. S1). DSSO contains two amine-reactive N-hydroxysuccinimide (NHS) ester chemical groups capable of covalently coupling to the terminal amines of lysine amino acid side chains. Based on the length of the cross-linker and the extended lengths of two lysine side chains, a DSSO cross-link provides direct evidence that two lysine residues are positioned nearby in space, and are within 30 Å from each other, as measured between their respective Cα atoms (Fig. 1a).

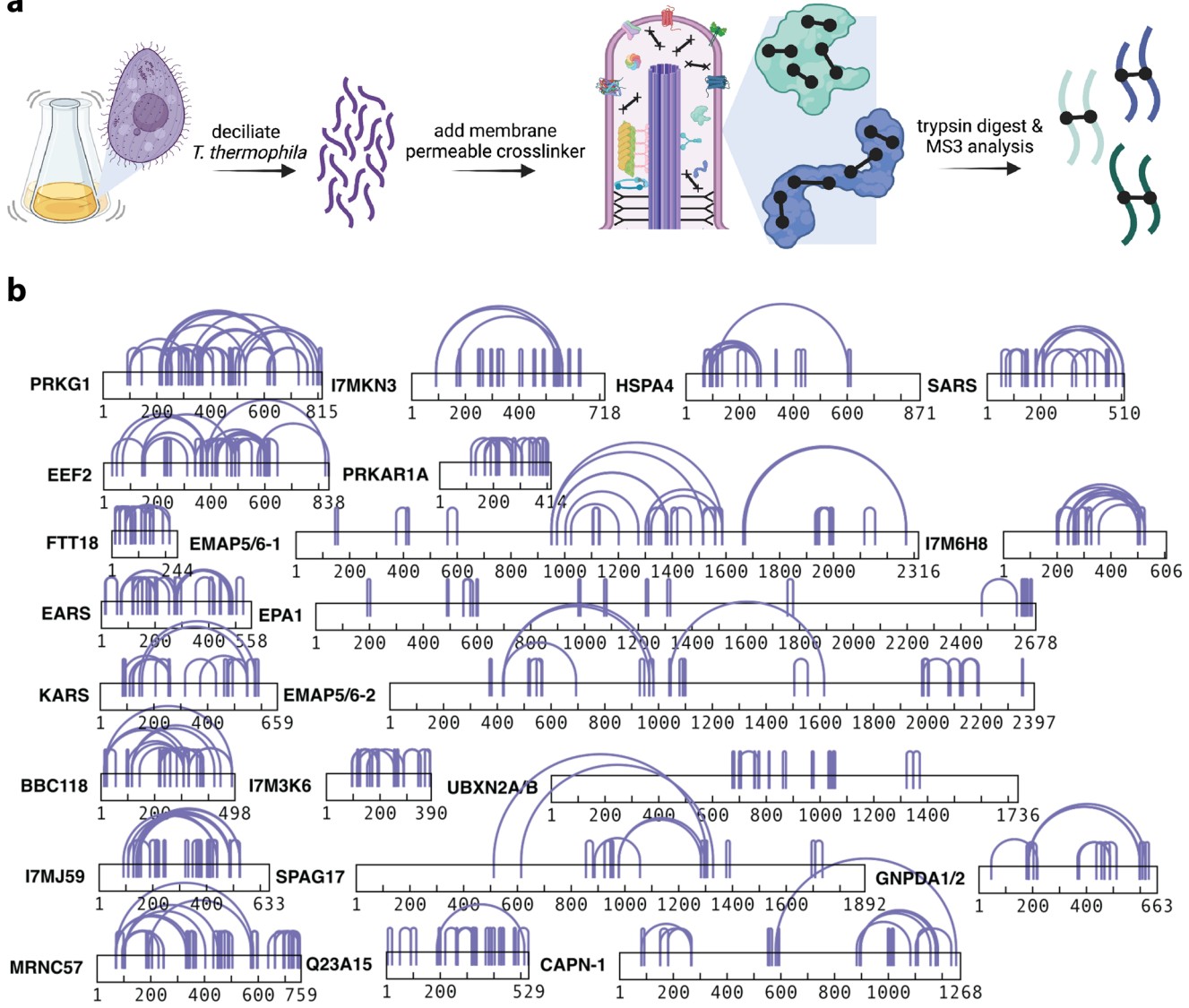

**Fig. 1 Chemical cross-linking of isolated ciliary proteins provides abundant intramolecular cross-links. a** Schematic of the protocol used to determine chemical cross-links among *Tetrahymena thermophila* ciliary proteins, from cell culture through ciliary isolation, incubation with the membrane-permeable cross-linker DSSO, to the use of tandem (MS$^1$/MS$^2$/MS$^3$) mass spectrometry to identify the specific cross-linked peptides. Created with BioRender.com. **b** Examples of the most extensively intramolecularly cross-linked proteins observed. The corresponding Uniprot identifiers and amino acid sequences are provided for all proteins discussed in the supporting Zenodo archive, along with the precise locations of the cross-links.

The coverage of the most abundantly intramolecular cross-linked proteins was extensive and spanned the full lengths of most sequences (Fig. 1b). To build confidence that our cross-link data do indeed faithfully capture biological protein structures, we compared our intramolecular cross-links against available experimental cryo-EM structures determined for the *T. thermophila* outer dynein arms (ODA) proteins[41]. For the three dynein heavy chains, comprising 13,382 amino acids in all, we observed a total of 155 intramolecular cross-links (Fig. 2a). This large number of intramolecular cross-links may be explained by the enrichment of dynein heavy chains in the endogenous sample and high-concentration of dynein heavy chains in the in situ experiment. After removing cross-links that occurred in regions without known structure, 124 cross-links could be positioned on the ODA structures (Fig. 2b). Across all three dyneins (Fig. 2c), 97% of the 124 cross-links were observed to connect lysines less than 30 Å apart (Cα-to-Cα), falling within the expected distance. However, the few violations we saw were quite large, with

distances greater than 200 Å. Due to these uncharacteristically large violations, we considered the possibility that these cross-linked pairs captured intermolecular contacts between adjacent copies of identical dynein proteins, reflecting the higher-order in situ arrangement of these proteins inside cilia.

To model the native arrangement of these dynein arms, we aligned the ODA structures into a subtomogram average determined from cryo-electron tomography of intact cilia axonemes[42] using ChimeraX[43]. Mapping cross-link violations onto the resulting assembly showed that the cross-links were now well-accommodated (i.e., less than 30 Å) by the dynein oligomer structure (Supplementary Table S1), boosting the agreement to 99% between the structure and the 124 cross-links, with the only violation being a single cross-link occurring at 34 Å, just above the maximum expected distance.

This near-perfect concordance between the experimental outer dynein arm structure and our cross-link dataset strongly supported the use of the cross-links to assess the quality of

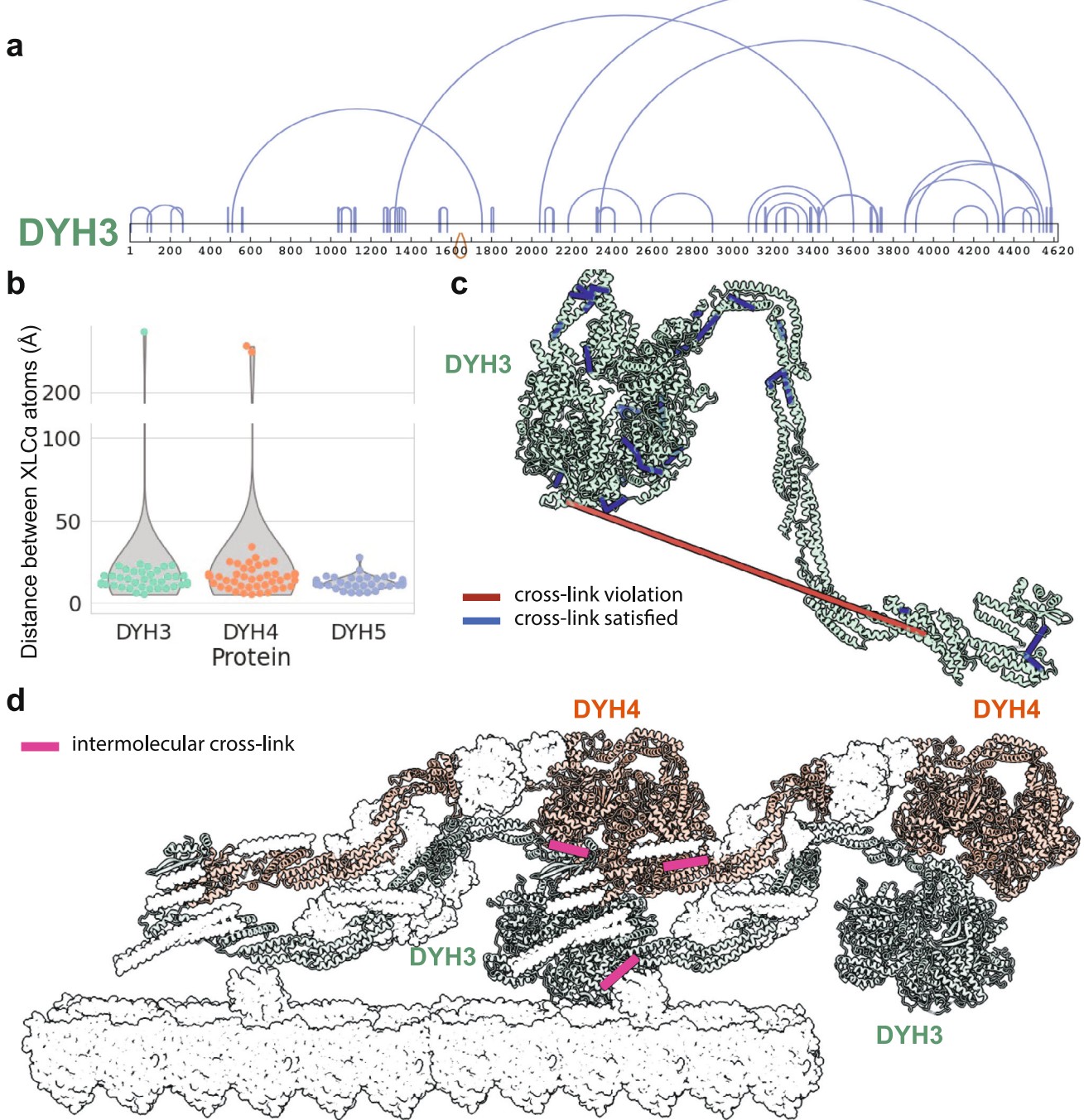

**Fig. 2 In situ cross-links agree with the known *T. thermophila* outer dynein arm cryo-EM structure. a** Cross-link diagram for DYH3 shows the abundance of intramolecular cross-links within the protein. **b** We observed a total of 155 intramolecular cross-links across all three dynein heavy chain proteins, 124 of which corresponded to structured regions and hence could be used as a validation set. Intramolecular distances are plotted for these 124 cross-links. **c** Intramolecular cross-links mapped onto the DYH3 structure. In summary, there was a 97% agreement between cross-links and cryo-EM structures of the dynein proteins. **d** In situ assembly of ODAs, show that perceived monomer cross-link violations are actually satisfied between copies of dynein proteins, improving the cross-link agreement to 99% (PDB ID:7MOQ) (see also Supplementary Table S1 for specific values).

AF2-predicted protein structures. We selected the 100 most highly cross-linked *Tetrahymena* ciliary proteins and predicted their structures using AF2 (Supplementary Table S2). Across this protein set, we had experimental measurements for a total of 1225 intramolecular cross-links, 86.2% of which agreed with the predicted structures. With longer distance thresholds of 35 and 40 Å, we measured 89.6% and 92.2% agreement,

respectively. Impressively, 43 predicted structures had no violations at all.

In order to gain some insight into areas of disagreement, we compared the number of cross-link violations per protein to the protein's average predicted local distance difference test (pLDDT) confidence score[9] (Fig. 3). Proteins with the most cross-link violations generally tended to have lower pLDDT scores,

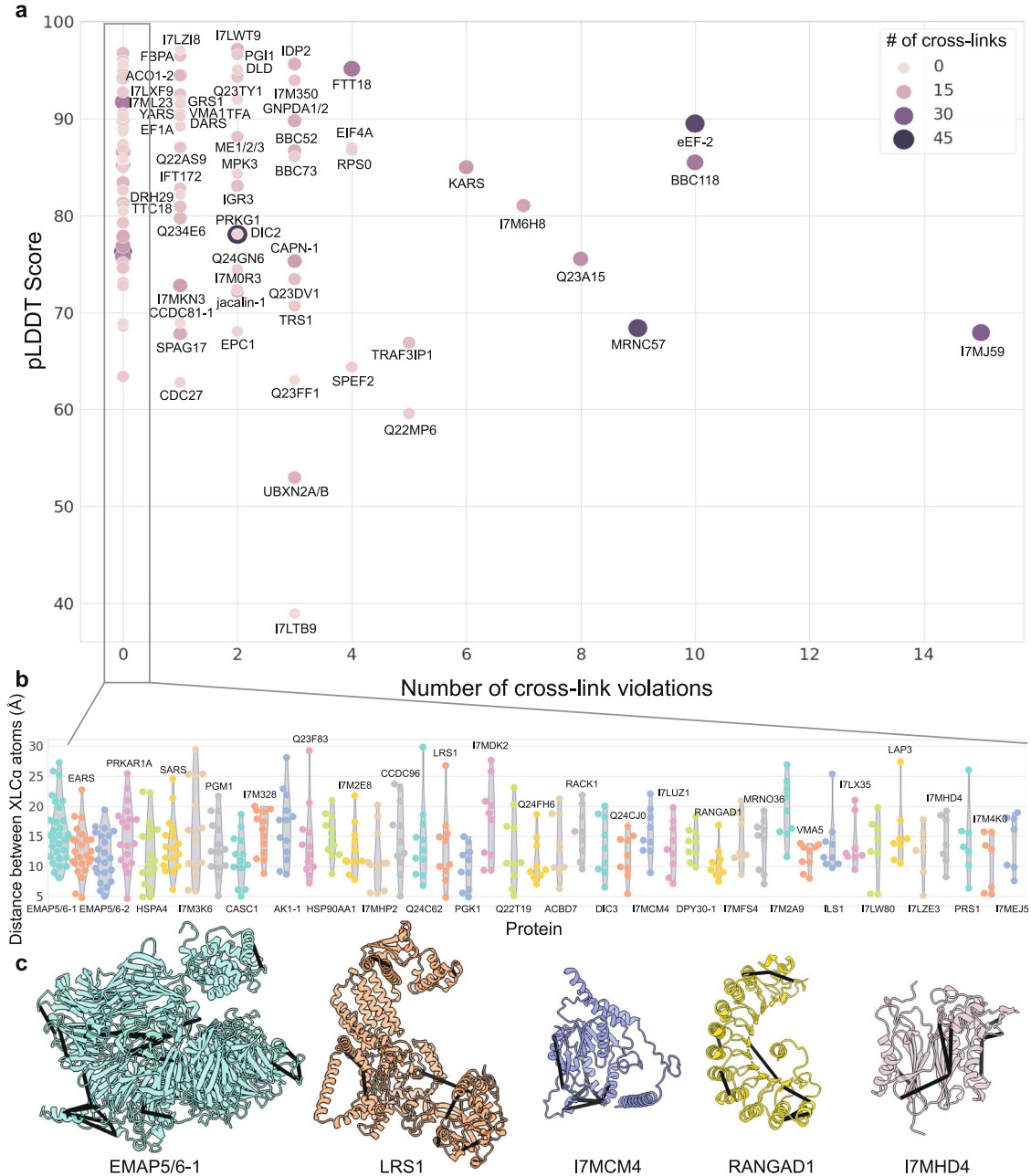

**Fig. 3 A general trend for fewer cross-link violations in AlphaFold2 models with higher pLDDT scores. a** Number of cross-link violations plotted against the pLDDT score for each of the *T. thermophila* proteins predicted. The size and shade of each dot represent the number of intramolecular cross-links for a given protein. The full data are provided as Supplementary Table S2. **b** A distance distribution view of the 43 proteins with no cross-link violations. **c** A selection of proteins from (**b**) with cross-links mapped onto the AF2-predicted structure.

consistent with AF2's reduced confidence in these predictions. Of the 13.8% of cross-link violations, about a third occurred in proteins with pLDDT scores below 70. The remainder of the violations occurred in reasonably confident protein structures (pLDDT over 70), leading us to further explore these regions within the predicted structures.

The predicted aligned error (PAE) scores produced by AF2 can be used to distinguish well-structured regions and well-folded domains within a protein structure from poorly predicted or unstructured regions[44]. PAE scores can thus be used to roughly define rigid domains or sets of domains within proteins that have the potential to be positioned in multiple orientations relative to each other. We, therefore, examined the PAE scores for our structures to determine if cross-link disagreements were more

likely to occur within or between these well-structured regions. By analyzing the PAE score maps using a watershed algorithm, we could segment the AF2-predicted structures to identify the best-predicted, contiguous, well-structured regions (Supplementary Fig. S2). We used this approach to examine the largest outliers in Fig. 3.

Among proteins with many cross-link violations, BBC118 stood out for having 10 violations despite a pLDDT score greater than 85, indicating a fairly confident structural prediction. To better understand why such a confidently predicted structure might have so many violations, we segmented its PAE score map to define well-predicted regions and asked whether the violations occur within or between these regions (Fig. 4a). Interestingly, these domains did not align exactly with

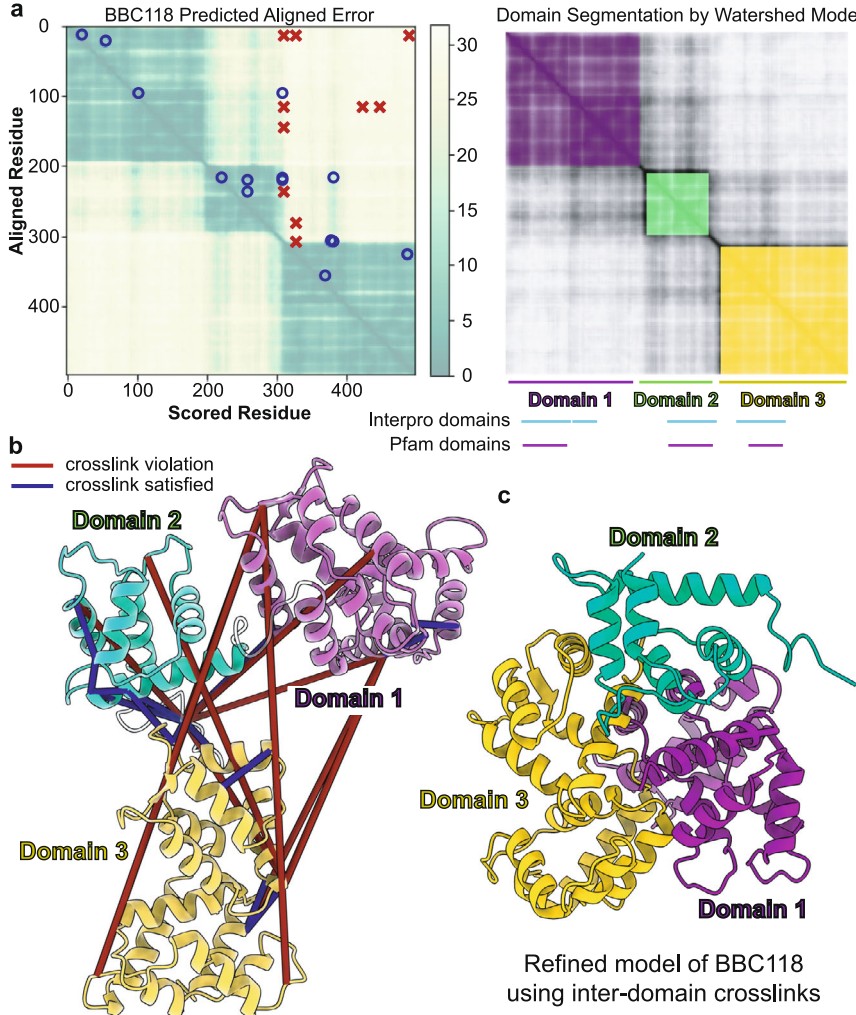

**Fig. 4 Cross-link violations tend to occur between or outside of structurally well determined regions. a** The predicted alignment error (PAE)[2] for *T. thermophila* protein BBC118 (Uniprot identifier I7ME23) with satisfied and violated cross-links plotted onto the heatmap. Blue circles are the satisfied cross-link and red x's are the cross-link violations. **b** We apply a watershed model to the PAE heatmap to segment the protein into individual rigid bodies. For BBC118, all cross-link violations occur between segmented rigid bodies. **c** The protein rigid bodies were broken up by the segmentation from the PAE and modeled using the intramolecular cross-links as distance restraints to find an arrangement that satisfied all but one of the cross-links. All models and PAE plots are provided at the supporting Zenodo web site.

known Pfam or InterPro domain annotations, which corresponded to the individual or grouped EF-hand motifs; in contrast, AF2 captured more extensive regions including interdomain segments whose structures could be confidently predicted. We plotted our intramolecular cross-links onto the PAE heatmap and onto the predicted structure (Fig. 4a, b) and found that all ten violations occurred between AF2 domains, while the eight cross-links falling within the domains satisfied the allowable cross-link distance. These results suggest that while AF2 may produce confident structure predictions locally within rigid bodies, there may be ambiguity in placing such rigid bodies relative to each other.

To test whether or not a conformation satisfying all cross-links was even possible, we divided the BBC118 3D structure into three rigid bodies based on the PAE segmentation boundaries, consisting of amino acids 8–195, 201–296, and 311–498, and we computed an integrative model[45] using the rigid bodies and 25 intramolecular cross-links. The resulting model of BBC118 satisfied all but one of the cross-links (Fig. 4c), showing that such a structural arrangement is physically plausible. Furthermore, when we similarly segmented (based on PAE) all 13

AF2-predicted structures with four or more cross-link violations, we found that 89.9% of the violations occurred between AF2-predicted well-folded regions. It is important to note that these violations may not necessarily represent incorrect structure prediction but rather could also point to the existence of an unknown stable interaction or homo-oligomer involving the proteins, such as we observed in Fig. 2.

Given the strong relationship between the PAE scores and the cross-link violations, we next examined this relationship more systematically. Binning the cross-linked amino acids across all 100 proteins according to their PAE scores (Fig. 5) revealed a linear relationship between the PAE and the cross-links, where larger PAE values correspond to a larger proportion of cross-link violations. The PAE range of 0–3.5 showed no cross-link violations, suggesting AF2's high confidence is appropriate in this regime. Overall, this analysis suggested that the PAE measure is reasonably well-calibrated and serves as an excellent indicator to help interpret the relative quality of specific regions of AF2 structures.

We observed one additional challenge for AF2: dynamic proteins that undergo large domain movements. In our data, this

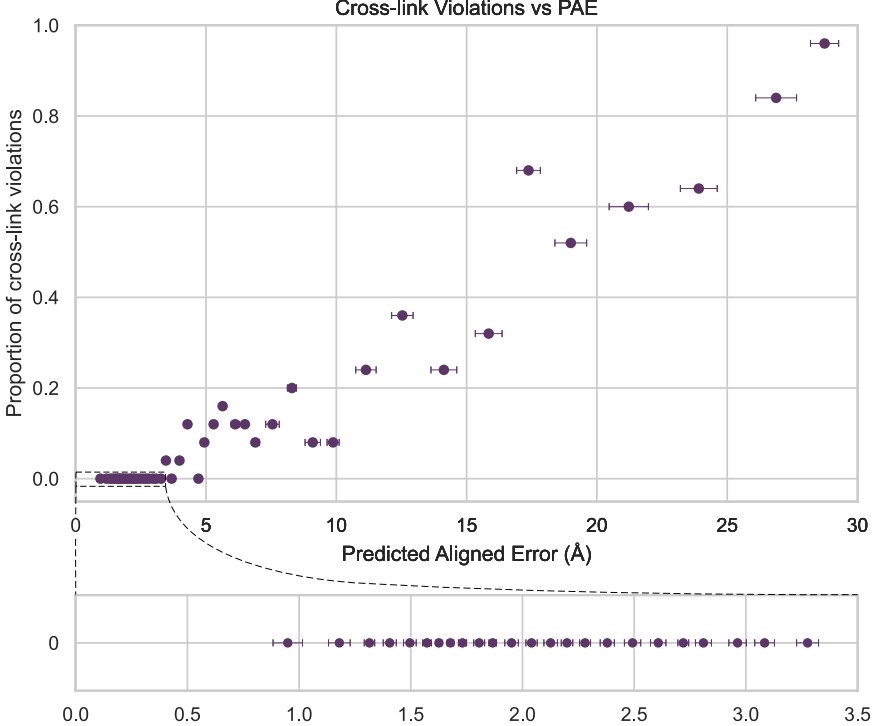

**Fig. 5 The proportion of cross-link violations is well-predicted by AF2's Predicted Aligned Error score, suggesting that it accurately captures the accuracy of structural models.** Considering the full set of cross-links in the 100 proteins, we ranked all cross-linked amino acid pairs by increasing PAE scores and divided them into 49 bins, comprising 25 cross-links per bin. For each bin of PAE values, we plotted the mean PAE score (+/− 1 standard deviation) and the proportion of in situ cross-links violated within that bin (in the unrelaxed AF2-predicted structures). All relevant data are located in the Zenodo repository, accompanied by a Python notebook to compute raw and average distances.

trend was evident for eEF-2, which, similar to BBC118, exhibited 10 cross-link violations despite an extremely confident pLDDT score (~90). Again, for eEF-2, the cross-link violations all occurred between compact domains. However, eEF-2 differed from BBC118 in that the regions between the domains had high per-residue pLDDT scores.

We investigated the role of these dynamics by first verifying that the structures predicted by each of the five AF2 models did not suggest any significant domain movements. Indeed, a comparison between these structures confirmed that all five predictions were highly similar, with the largest RMSD between structures being 1.01 Å. To investigate further, we examined the four experimentally determined structures of yeast orthologs[46–48] obtained from the Protein Data Bank[28] (Fig. 6). Each structure was determined with different binding partners, and collectively, they reveal that the two domains of yeast eEF-2 exhibit considerable conformational flexibility with respect to each other. Homology modeling of the *T. thermophila* protein onto each of the yeast ortholog structures reveals that the AF2-predicted structure shows another conformation of the two domains, distinct from the four other orientations; all five structures exhibit multiple cross-link violations, suggesting that the *T. thermophila* eEF-2 likely samples multiple conformations inside the cilia. Regardless, the AF2 structure prediction, while largely correct for the separate domains, fails to capture the dynamics of their relative positions for this protein.

**Conclusions**. In this paper, we used chemical cross-linking and mass spectrometry of *T. thermophila* ciliary proteins to interrogate their 3D structures within their native contexts and assemblies. These data, in turn, allowed us to evaluate the concordance of AlphaFold2's predictions of these protein structures. Impressively, 43% of AF2-predicted protein structures show no

disagreements with the in situ cross-links, and a large majority (87%) showed three or fewer cross-link violations, demonstrating AF2 predicts biologically relevant protein conformations.

However, our study also highlights the importance of experimental validation. For specific cases, high-confidence structures were predicted that exhibited a number of cross-link violations, 89.9% of which fell outside well-predicted domains or in unstructured segments. Multidomain proteins can exhibit varying arrangements of their rigid body domains, which can pose a significant challenge for AlphaFold. Importantly, we confirm that the PAE scores provide useful guidance for defining these domain boundaries.

Overall, this combination of AF2 and cross-linking data can add confidence to the models, guide their interpretation, and may also serve as a valuable complement to other approaches, such as cryo-electron tomography, for elucidating proteins' endogenous structures.

## Methods

***T. thermophila* culture**. *Tetrahymena thermophila* SB715 were obtained from the *Tetrahymena* Stock Center (Cornell University, Ithaca, NY) and maintained in Modified Neff medium obtained from the stock center at room temperature (~21 °C). To prepare cilia, 10 ml cultures were expanded at 30 °C with shaking (100 rpm) directly before cilia isolation.

**Deciliation of *T. thermophila* and in situ cross-linking**. *Tetrahymena* were resuspended in Hepes Cilia Wash Buffer (H-CWB) [50 mM HEPES pH 7.4, 3 mM MgS0$_4$, 0.1 mM EGTA, 250 mM sucrose, 1 mM DTT, 1× Complete protease cocktail, 1× PhosSTOP cocktail]. Intact cilia were released by dibucaine treatment[40], and all subsequent steps were performed at 4 °C. After removing cells and debris, cilia were recovered by centrifugation (17,000 × *g*, 5 min), and washed once in H-CWB. A cilia pellet of ~10 μl was resuspended in 50 μl H-CWB. Cross-linking was performed by the addition of 5 μl DSSO stock (freshly made 50 mM in anhydrous DMSO) to 5 mM final concentration and incubation for 1 h at room temperature. Cross-linking was quenched by adding 1 M Tris pH 8.0 to 33 mM for 30 min at room temperature.

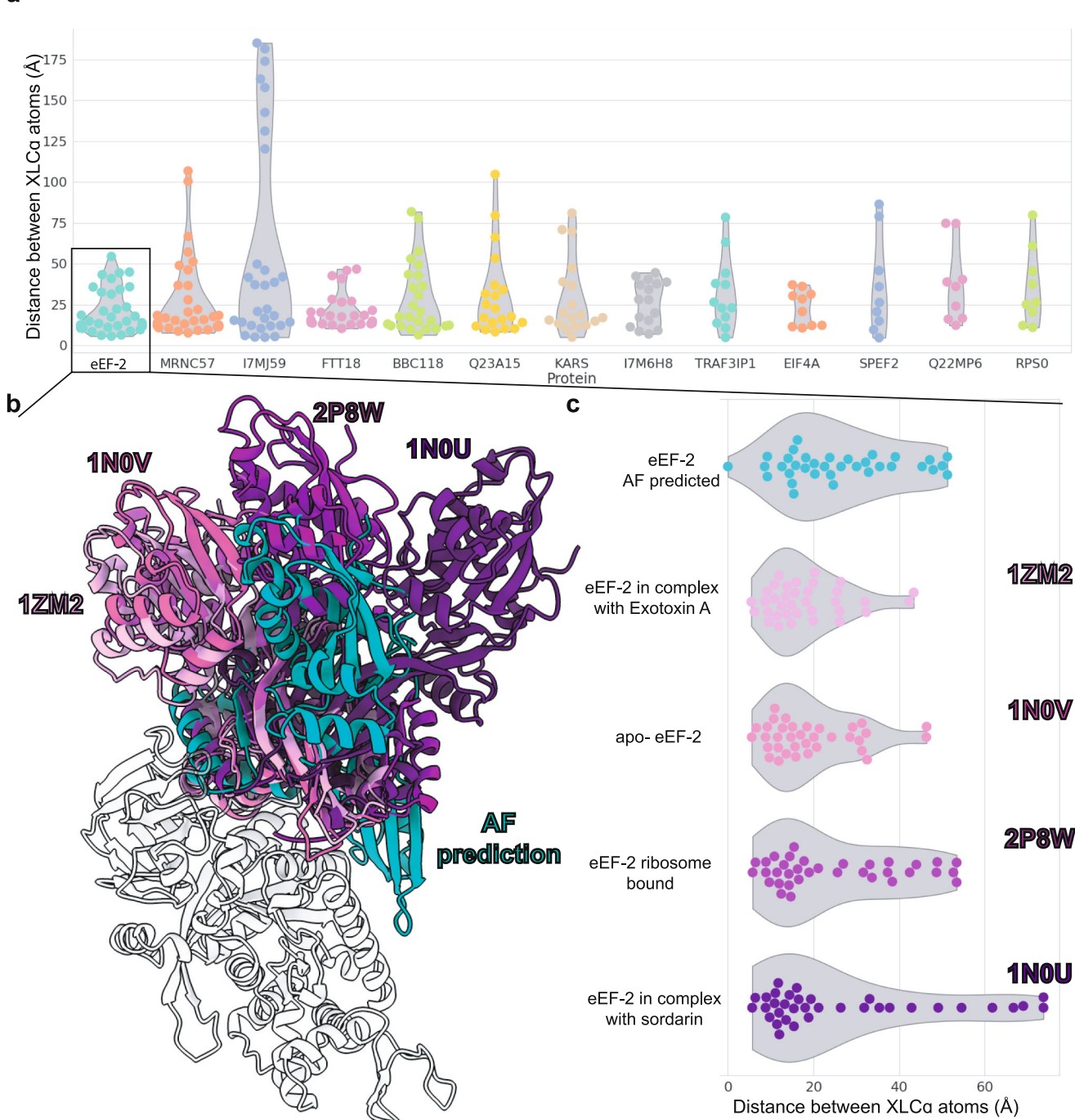

**Fig. 6 Predictions for the protein eEF-2 show that the AF2 model differs from 4 homologous crystal structures and the cross-links due to interdomain rearrangements. a** Distribution of cross-link distances for proteins in our dataset with four or more cross-link violations (data are available on the supporting Zenodo site). **b** A hinge-like motion is evident between the two domains of the AF2 structure of the *T. thermophila* eEF-2 protein (Uniprot accession Q22DR0)(cyan) compared to four eEF-2 structures solved by X-ray crystallography and showing structures determined in the presence of different binding partners[46–48]. All structures were superimposed on the N-terminal GTP binding domain. **c** indicates the distribution of 35 cross-link distances in each structure, with the appropriate PDB identifiers labeled to the right of the violin plots.

The cross-linked sample was prepared for mass spectrometry as in[49]. Specifically, cross-linked cilia were solubilized in 2% SDS at 95 °C, and proteins were precipitated by adding six volumes of acetone, incubated overnight at 4 °C, and precipitated protein was collected by centrifugation at $13,000 \times g$ 4 °C for 15 min. The protein pellet was washed with acetone twice, dried, resuspended in 200 µl 1% sodium deoxycholate/50 mM $NH_4HCO_3$, and sonicated ($2 \times 10$ min.) in a water bath. Proteins were reduced with 5 mM TCEP at 56 °C for 45 min, alkylated with 25 mM iodoacetamide in the dark for 45 min, quenched with 12 mM DTT, then digested overnight with 2 µg trypsin in 1 ml final volume at 37 °C. Digestion was stopped by the addition of formic acid to 1%, and the deoxycholate precipitate

was removed by centrifugation at $16,000 \times g$ for 10 min. The supernatant volume was reduced in a vacuum centrifuge, and peptides were filtered through a 10,000 MWCO Amicon Ultra 0.5 ml device (Millipore) before desalting with a C18 spin tip (Thermo Scientific HyperSep SpinTip P-20 BioBasic # 60109-412) as in ref.[50]. To enrich for cross-linked peptides, the desalted peptides were dried and resuspended in 50 µl 30% acetonitrile, 0.1% TFA, and separated on a GE Superdex 30 Increase 3.2/300 size-exclusion column (Cytiva) at 50 µl/minute flow rate using an ÄKTA Pure 25 FPLC chromatography system (Cytiva). 100 µl fractions were collected, dried, and resuspended in 5% acetonitrile, 0.1% formic acid for mass spectrometry.

**Mass spectrometry**. Mass spectra were collected on a Thermo Orbitrap Fusion Lumos tribrid mass spectrometer as follows: Peptides were separated using reverse phase chromatography on a Dionex Ultimate 3000 RSLCnano UHPLC system (Thermo Scientific) with a C18 trap to Acclaim C18 PepMap RSLC column (Dionex; Thermo Scientific) configuration. An aliquot of the cross-linked peptides prior to SEC enrichment was analyzed using a standard top-speed HCD MS1-MS2 method[51] and analyzed using the Proteome Discoverer basic workflow. Proteins identified were exported as a fasta file to serve as the look-up database for cross-link identification in the cross-link-enriched fractions. For identification of DSSO cross-links, spectra were collected as follows: peptides were resolved using a 115 min 3–42% acetonitrile gradient in 0.1% formic acid. The top-speed method collected full precursor ion scans (MS1) in the Orbitrap at 120,000 $m/z$ resolution for peptides of charge 4–8 and with dynamic exclusion of 60 s after selecting once, and a cycle time of 5 s. CID dissociation (25% energy 10 msec) of the cross-linker was followed by MS2 scans collected in the orbitrap at 30,000 $m/z$ resolution for charge states 2–6 using an isolation window of 1.6. Peptide pairs with a targeted mass difference of 31.9721 were selected for HCD (30% energy) and collection of rapid scan rate centroid MS3 spectra in the Ion Trap. Data were analyzed using the XlinkX node of Proteome Discoverer 2.3 and the XlinkX_Cleavable processing and consensus workflows, selecting cross-links with a False Discovery Rate of 1%, and results were exported to xiView[52] for visualization.

We supplemented these in situ cross-links with additional cross-links collected from native (non-denaturing) protein extracts from isolated *Tetrahymena* cilia prepared as above. These data were previously collected and analyzed by mass spectrometry (available from the MassIVE database under accession ID MSV000089131) as described in ref. [25], which focused solely on the analysis of the Intraflagellar Transport A protein complex. For this work, we considered all intramolecular protein cross-links captured by these data, analyzed identically as for the in situ data. A comparison of the in situ and native extract cross-link sets showed high concordance (Supplementary Fig. S1), and we therefore performed all tests using the union of the two sets.

**AlphaFold2 structure prediction**. We sorted the identified ciliary proteins by decreasing counts of intramolecular chemical cross-links per protein and selected the top 100 proteins with the most intramolecular cross-links to serve as a test set for structure prediction and subsequent analyses. Protein structures were predicted using the 2.1.2 version/release of AlphaFold2[2] as implemented on Texas Advanced Computing Center (TACC) Maverick2 and Frontera[53] GPU computer clusters. Structures were predicted using the monomer and predicted template modeling (pTM) AF2 protocols[2] and are available for download on the supporting Zenodo data repository.

We selected the unrelaxed predicted structures from monomer model 1 in order to increase throughput and remain within the allocated maximum time limits for the TACC clusters. Proteins with the most cross-link observations were selected as candidates for AF2 prediction, and then the top 100 proteins for which completed unrelaxed structure predictions could be derived were selected for further analysis. To confirm there was no significant variation in cross-link violation between unrelaxed and relaxed AF2 predictions, we also predicted relaxed structures for the top ten most cross-linked proteins (Supplementary Fig. S3). There were no differences in cross-link agreement between these unrelaxed and relaxed predictions. In addition, we judged whether limiting our predictions to only model 1 might affect our results by predicting all 5 monomer models for one example protein, eEF-2.

**AlphaFold2 domain boundary prediction**. To identify whether violated cross-links occurred within or between domains, we identified proteins that had four or more cross-link violations (a total of 13 proteins) for the predictions' pLDDT scores (based on the unrelaxed structure). The AF2 monomer pTM model 1 was used to predict these protein structures again with predicted aligned error (PAE) scores. PAE is calculated for two residues $x$ and $y$ as the predicted error (in units of Angstroms (Å)) for the position of $x$ when assuming that the predicted position for $y$ is correct. Regions of low PAE were used to identify well-structured domains or well-predicted regions using segmentation of a PAE matrix.

PAE scores are not symmetrical, but we were interested in using low error regions to analyze distributions of recorded cross-links, which are symmetrical between residues. Therefore, to incorporate the information from both directions of the PAE scores, the PAE matrix was averaged with its transpose to create a matrix symmetrical across its diagonal. We then denoised the matrix one or more times using a median filter, and applied a gradient filter to generate the topography for watershed segmentation. Initial basin markers were defined where another gradient filter found values below a chosen threshold. The gradient was then used as the input to the watershed segmentation transform from scikit-image[54], along with the identified markers, to produce a segmented version of the PAE heatmap.

Due to the low PAE values of one residue compared to itself or its close neighbors, thin segments were often identified along the diagonal of the image. Since these thin segments do not represent regions that are fully interconnected with low error, we removed labels on areas not meeting a minimum width threshold, and any gaps created by this process within a region were filled with that region's label. Finally, labels were assigned to each residue by traversing the

diagonal of the resulting segmented matrix. The number of times the denoise filter was applied, and the window size of each filter was configured per protein to produce labels that appeared to match the PAE heatmap well.

Code for the watershed-segmentation approach for identifying well-predicted regions is provided on the supporting Zenodo repository.

**BBC118 modeling**. We used the Integrative Modeling Platform[45] to refine the 3D structural model of BBC118 taking into account our cross-links as distance restraints. We used the AF2 prediction of the structure as the initial state of the protein in our modeling. BBC118 was modeled as a chain of rigid bodies, where the boundaries of the rigid bodies were defined using our watershed segmentation of the output AF2 PAE heatmap. DSSO intramolecular cross-links were used as cross-linking restraints in the modeling, which was run for a total of 20,000 frames from 10 random initial conditions. Model convergence and sampling exhaustiveness were assessed using standard methods[55]. A total of 200,000 models were produced with 29,000 models in the final model cluster, which had a cluster precision of 3.369 Å, and which markedly improved the agreement with cross-linking data while maintaining reasonable packing of domains (Supplementary Fig. S4).

**Statistics and reproducibility**. Cross-links were selected for this study at a False Discovery Rate of 1%. We predicted the structures for only the top 100 best-sampled proteins from our XL/MS data. Summary statistics for the AF2-predicted structures are available in the supporting Zenodo repository. For our improved model of BBC118, all statistics are summarized in Supplementary Fig. S4.

**Reporting summary**. Further information on research design is available in the Nature Portfolio Reporting Summary linked to this article.

## Data availability

Mass spectrometry proteomics data were deposited in the MassIVE/ProteomeXchange databases (https://massive.ucsd.edu, see also ref. [56])) under MassIVE accession numbers MSV000089917 and MSV000090056. Additional supporting materials, including all AF2 3D models, are available in a supporting Zenodo repository available at https://doi.org/10.5281/zenodo.7725518.

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

## Acknowledgements

The authors gratefully acknowledge the generous support of the *Tetrahymena* stock center (Cornell University) that made this project possible and thank John Jumper (DeepMind) for helpful suggestions. Research was funded by grants from the National Institute of General Medical Sciences R35GM122480 (to E.M.M.) and R35GM138348 (to D.W.T.), National Science Foundation (2019238253 to C.L.M.), National Institute of Child Health and Human Development (HD085901 to E.M.M.), and Welch Foundation (F-1515 to E.M.M., F-1938 to D.W.T.). D.W.T. is a CPRIT Scholar supported by Cancer Prevention and Research Institute of Texas (RR160088). The authors thank the Texas Advanced Computing Center at The University of Texas at Austin for providing high-performance computing resources that have contributed to the research results reported in this paper.

## Author contributions

Conceptualization: C.L.M., E.L.P., O.P., and E.M.M.; formal analysis: C.L.M. and E.L.P.; investigation: C.L.M., E.L.P., and O.P.; methodology: C.L.M., E.L.P., and O.P.; funding and resources: C.L.M., D.W.T., and E.M.M.; writing—original draft: C.L.M., E.L.P., O.P., and E.M.M.; writing—review and editing: C.L.M., E.L.P., O.P., D.W.T., and E.M.M.

## Competing interests

The authors declare no competing interests.
