## [Peer Review File · Communications Biology]

Reviewers' comments:

Reviewer #1 (Remarks to the Author):

The submitted work from McCafferty et. al. is a nice validation of predicted AlphaFold2 model proteins using cross-linking mass spectrometry. I have few comments which might of help to increase clarity.

A. Page 4, Last paragraph - These results present how well AF2 models work and could be presented in a table.

B. Figure S2 - Colors in fig K could have more contrast for clarity.

Reviewer #2 (Remarks to the Author):

Authors Caitlyn L. McCafferty et al. in this manuscript have investigated whether AlphaFold2 structure predictions match native protein conformations using the cross-linking mass spectrometry of endogenous ciliary proteins for experimental validation. Authors have performed chemical cross-linking of ciliary proteins in situ within intact *Tetrahymena thermophila* and native enriched extracts using the membrane-permeable, mass-spectrometry cleavable chemical cross-linker disuccinimidyl sulfoxide (DSSO). Authors have found 1,225 intramolecular cross-links within the 100 best-sampled proteins to benchmark the distance restraints obeyed by proteins in their native assemblies.

Using this protein set, authors have achieved good concordance of about 86.2% between our cross-links and AF2 structure predictions, with some disagreements or violations between domains of multi-domain proteins and dynamic proteins of conformational change. Authors have shown that 13.8% of cross-link violations occurred in Proteins with the average predicted local distance difference test (pLDDT) confidence score below 70. The remainder of the violations also occurred in reasonably confident protein structures (pLDDT over 70). Here, authors have come up with a unique calibration between predicted aligned error (PAE) scores produced by AF2 and the experimentally obtained chemical-cross link violations, by binning the cross-linked amino acids across all 100 proteins according to their PAE scores which established a linear relationship between the PAE and the cross-links, with larger PAE values correspond to a larger proportion of cross-link violations. Authors have showed AF2's high confidence is appropriate in this regime with no crosslink violations in the PAE range of 0 to 3.5.

Overall, the manuscript, and the detailed systematic analysis shows promising results for investigating more case studies and challenges posed by AF2 prediction and interpret the relative quality of specific regions of AF2 structures in native conformations.

Minor comments

- The title "Does AlphaFold2 model proteins' intracellular conformations?" takes me to the question raised from the following statement in page 4, "Impressively, 43% of AF2 predicted protein structures show no disagreements with the in situ cross-links, and a large majority (87%) showed three or fewer cross-link violations, demonstrating AF2 predicts biologically relevant protein conformations.

Here, it is not clear whether agreement between in situ-crosslinks and AF2 prediction is the governing factor for structure that's biologically relevant protein conformation? Or the AF2 prediction as it incorporates information from evolutionary coupling and amino acid conservation which should, in principle, capture structures most relevant to the predominant cellular roles of these proteins.

- Also, the authors have not discussed about any other chemical cross-linker molecule other than

DSSO. Is there any other mass spectrometry cleavable linker that can be used for this study? If so, does it alter the cross-linking coverage or violations

- Is there a concentration dependence for cross-linker in case higher oligomers or assembly protein such as outer dynein arms (ODA) proteins.

Reviewer #3 (Remarks to the Author):

The submitted manuscript describes efforts to validate AlphaFold2's prediction of endogenous ciliary proteins using chemical crosslinking coupled with MS methodology, as determined crosslink distances will be used to test the predicted models. The paper is well written, but I found the overall hypothesis to be fatally flawed. The authors state "Global benchmarking and independent validation of such predicted structures will be necessary to inform reliable and nuanced interpretations of these structures" and posit crosslinking to supply such benchmarking. The distances found in XLMS studies often do not correspond with known structures (for reviews, see Graziadei and Rappsilber, Structure 2021 and Mintseris and Gygi, PNAS 2019). While XLMS is a good complement, it is not yet appropriate for benchmarking purposes and cannot be used for experiment validation. A paper reporting the structure using predictive methods and XLMS together to predict the structure of endogenous ciliary proteins of unknown overall structure would be more favorably reviewed, but it appears that the focus of this paper greatly overstates the validation aspects. As such, I find the paper as written to be unacceptable for publication.

Responses to referees

McCafferty *et al.*, Does AlphaFold2 model proteins' intracellular conformations? An experimental test using cross-linking mass spectrometry of endogenous ciliary proteins

Our responses follow in-line in blue text

Reviewers' comments:

Reviewer #1 (Remarks to the Author):

The submitted work from McCafferty *et. al.* is a nice validation of predicted AlphaFold2 model proteins using cross-linking mass spectrometry. I have few comments which might of help to increase clarity.

A. Page 4, Last paragraph - These results present how well AF2 models work and could be presented in a table.

We thank the reviewer for this comment and have now incorporated the data from the final paragraph into Table S2.

B. Figure S2 - Colors in fig K could have more contrast for clarity.

We have now enhanced the contrast of the colors in Figure S2.

Reviewer #2 (Remarks to the Author):

Authors Caitlyn L. McCafferty *et al.* in this manuscript have investigated whether AlphaFold2 structure predictions match native protein conformations using the cross-linking mass spectrometry of endogenous ciliary proteins for experimental validation. Authors have performed chemical cross-linking of ciliary proteins in situ within intact *Tetrahymena thermophila* and native enriched extracts using the membrane-permeable, mass-spectrometry cleavable chemical cross-linker disuccinimidyl sulfoxide (DSSO). Authors have found 1,225 intramolecular cross-links within the 100 best-sampled proteins to benchmark the distance restraints obeyed by proteins in their native assemblies.

Using this protein set, authors have achieved good concordance of about 86.2% between our cross-links and AF2 structure predictions, with some disagreements or violations between domains of multi-domain proteins and dynamic proteins of conformational change. Authors have shown that 13.8% of cross-link violations occurred in Proteins with the average predicted local distance difference test (pLDDT) confidence score below 70. The remainder of the violations also occurred in reasonably confident protein structures (pLDDT over 70). Here, authors have come up with a unique calibration between predicted aligned error (PAE) scores produced by

AF2 and the experimentally obtained chemical-cross link violations, by binning the cross-linked amino acids across all 100 proteins according to their PAE scores which established a linear relationship between the PAE and the cross-links, with larger PAE values correspond to a larger proportion of cross-link violations. Authors have showed AF2's high confidence is appropriate in this regime with no crosslink violations in the PAE range of 0 to 3.5.

Overall, the manuscript, and the detailed systematic analysis shows promising results for investigating more case studies and challenges posed by AF2 prediction and interpret the relative quality of specific regions of AF2 structures in native conformations.

We thank the referee for this excellent distillation of the paper.

Minor comments

- The title “Does AlphaFold2 model proteins’ intracellular conformations?” takes me to the question raised from the following statement in page 4, “Impressively, 43% of AF2 predicted protein structures show no disagreements with the in situ cross-links, and a large majority (87%) showed three or fewer cross-link violations, demonstrating AF2 predicts biologically relevant protein conformations.

Here, it is not clear whether agreement between in situ-crosslinks and AF2 prediction is the governing factor for structure that’s biologically relevant protein conformation? Or the AF2 prediction as it incorporates information from evolutionary coupling and amino acid conservation which should, in principle, capture structures most relevant to the predominant cellular roles of these proteins.

We thank the reviewer for raising this question. We used our *in situ* XL/MS data as a direct experimental validation of amino acid positions because it can capture protein arrangements as they exist in the cell (including dynamics). We therefore consider the agreement between AF2 prediction and our XL/MS data as a validation of AF2’s ability to produce biologically relevant conformations of proteins. As to how AF2 arrives at these structures, it is indeed likely that at least part of AF2’s ability to capture these biological conformations may derive from the evolutionary coupling data used in building the models. We now emphasize this point on p. 3 and again on p. 4.

- Also, the authors have not discussed about any other chemical cross-linker molecule other than DSSO. Is there any other mass spectrometry cleavable linker that can be used for this study? If so, does it alter the cross-linking coverage or violations

Other cross-linkers are indeed available that might be used (although their application would be beyond the scope of our goals here, which was to assess AlphaFold-predicted structures using independently obtained experimental protein structure data for a large number of proteins). To address this comment, as well as comments by Reviewer 3 below, we now add a section to the paper on p. 3 with discussion and literature review of other cross-linkers and their applications to structure validation.

- Is there a concentration dependence for cross-linker in case higher oligomers or assembly protein such as outer dynein arms (ODA) proteins.

This is an interesting question for which we have no simple answer. There is definitely a strong concentration dependence to which sites are cross-linked, and having many replicate copies of the same interface certainly increases the chances of observing the cross-links. However, it is not clear that higher order assemblies should be any more likely to be cross-linked than the same number of e.g. dimers in solution. It is more likely the concentration of the interfaces that matters for these assays, not the overall oligomeric state of the assemblies, unless the cross-linkers themselves should e.g. promote oligomerization. We have seen no evidence of this. We now briefly note the effect of concentration on p. 4.

Reviewer #3 (Remarks to the Author):

The submitted manuscript describes efforts to validate AlphaFold2's prediction of endogenous ciliary proteins using chemical crosslinking coupled with MS methodology, as determined crosslink distances will be used to test the predicted models. The paper is well written, but I found the overall hypothesis to be fatally flawed. The authors state "Global benchmarking and independent validation of such predicted structures will be necessary to inform reliable and nuanced interpretations of these structures" and posits crosslinking to supply such benchmarking. The distances found in XLMS studies often does not correspond with known structures (for reviews, see Graziadei and Rappsilber, Structure 2021 and Mintseris and Gygi, PNAS 2019). While XLMS is a good complement, it is not yet appropriate for benchmarking purposes and cannot be used for experiment validation. A paper reporting the structure using predictive methods and XLMS together to predict the structure of endogenous ciliary proteins of unknown overall structure would be more favorably reviewed, but it appears that the focus of this paper greatly overstates the validation aspects. As such, I find the paper as written to be unacceptable for publication.

The referee makes quite a strong statement here ("I found the overall hypothesis to be fatally flawed"), and we wish to be quite sure that we understand correctly what is being objected to.

If the referee means that they believe that it is not possible to use XL/MS to measure *specific* distances between alpha carbons (*i.e.* to state that two residues must be precisely 30 Angstroms apart based on the cross-links), then we agree completely, and please note this is not what we are doing. Just in case this is what the referee was indeed referring to, we have now clarified the language on p. 4 to help other readers to avoid this interpretation.

Rather, we treat each crosslink as a distance *restraint*, *i.e.* a specific threshold distance that should not generally be exceeded. Chemical crosslinks correspond to actual physical entities with known atomic structures of fixed maximal lengths connecting pairs of amino acid residues. Leaving aside for a moment the issues of assignment errors, the direct experimental

observation of a crosslink implies that those two amino acids *must* have resided within the threshold distance for sufficient time and in a sufficient proportion of proteins in the population to be *robustly* chemically coupled and subsequently detected by mass spectrometry, implying that e.g. more than 100 million copies of the protein(s) were observed to adopt the necessary conformational state to be cross-linked, based on a typical (~fmol) detector sensitivity. As this is a very high bar in general to clear, the method shows strong abundance biases and preferences for long-lived, stable 3D protein conformations.

However, the referee's subsequent statement that "XLMS ... is not yet appropriate for benchmarking purposes and cannot be used for experiment validation" is at odds with a decade of research and a large and growing number of successful use cases across the field.

This is such a fundamental point that it is worth elaborating on.

Importantly, crosslinks are not predictions, but *independent experimental observations*. As with any such set of physical measurements, there will be some proportion of measurement and attribution errors, and protein dynamics will affect proximity as well. However, these effects have been well-studied in the literature, and crosslinks are widely appreciated across the field as important experimental validations of protein models. For example, Brodie *et al.*, in *Science Advances* 3(7):e1700479 (2017), explicitly state:

"Long-distance Lys-Lys cross-linking using amine-reactive reagents with spacer lengths of >10 Å ... can be used to validate [predicted] protein structures".

Similarly, Yu and Huang note in *Anal. Chem.* 90(1):144–165 (2018):

"cross-links can serve as direct evidence for pairwise protein interactions without further biochemical confirmation, as often required for conventional AP-MS experiments" and "cross-links can be utilized as distance constraints for various applications ... [including] structure validation and integrative modeling."

In fact, the two reviews cited by the referee explicitly support this use case:

(1) Graziadei and Rappsilber, *Structure* 2022, state:

"The capacity of crosslinking MS to provide distance restraints in complex samples, or even in cells, means that it has proven to be an ideal complementary technique for validating EM or crystallographic models in their native context (Casañal *et al.*, 2019; O'Reilly and Rappsilber, 2018; Schmidt and Urlaub, 2017)."

As this is quite a recent review (Jan 2022), the authors even note the potential for our specific application:

“Recently, there have been breakthroughs in protein structure prediction by the deep learning algorithms AlphaFold2 (Jumper et al., 2021; Tunyasuvunakool et al., 2021) and RosettaFold (Baek et al., 2021). Crosslinking-MS may be used to understand the arrangement of multidomain proteins and validate or select between proteins with different arrangements of low-confidence regions.”

They conclude that crosslinks

“can be used to validate models, map binding interfaces, or derive new models by docking and integrative modeling.”

(2) Mintseris and Gygi, *PNAS* 2019, perform their own comparison of structure and cross-links and find excellent concordance, including for the BS3 crosslinker, the closest match to our DSSO crosslinker, noting that:

“cross-linking data could be used to validate candidate cryo-EM models.”

The authors compute a structure-based FDR on their crosslinks (see Fig. S4), from which they

“estimate a value of 3.2% structure-based FDR for the combined dataset with a 95% confidence interval (2.0 and 4.5%)”.

That is to say, they observe a 97% concordance with independently-derived experimental structures.

As to the issue of protein dynamics, we refer the referee in particular to the study of Merkley *et al.*, *Protein Science* 23:747—759 (2014), whose study of both experimental and modeled structures lead them to conclude that even with the consideration of protein dynamics, that:

“~84% of the observed intramolecular crosslinks and ~86% of the observed intermolecular crosslinks have C α distances smaller than the theoretical (i.e., both linker and side chains fully extended) C α –C α distance of 24 Å, suggesting that the common 24-Å criterion is in fairly close agreement with the data” and that “Our estimated maximum crosslinking distance values (24–30 Å between C α atoms) agree well with the experimental distance distribution. For C α –C α distances, 89.3% of the observed crosslinks fall below our recommended maximum threshold of ~30 Å. Our recommended C α threshold value is similar to the values applied by Aebersold and coworkers (30 Å) and only slightly longer than the value applied by Sinz and coworkers (27.4 Å). Rappsilber and coworkers have used 27.4 Å, but more recently they suggest a range (25–29 Å). Thus, the Dymomeomics-based maximum C α recommendation is in line with the empirically determined threshold values already in common use.”

Importantly, all of these authors describe a significant concordance across the field for the use of crosslinks as experimental distance restraints and for independent structural validation, as well as for the specific 30 Angstrom threshold value we employ.

We have therefore taken the referee's comments to indicate that we should do a better job at communicating this well-established utility in the revised paper and at citing other examples using XL/MS to independently validate models derived by other approaches. To address these points, we now add the above references and include a condensed discussion of these points on p. 3.

REVIEWERS' COMMENTS:

Reviewer #3 (Remarks to the Author):

I appreciate the changes made by the authors, as well as the care they have taken to respond to my initial review. I do consider that we have a common interest in utilizing MS crosslinking restraints for integrative structural biology. My concerns were pointedly not to question the validity of measuring the outer bound of distance restraints in native crosslinking studies, and we cited common reviews showing the validity of crosslinking MS and its ability to complement other high resolution studies. I raised no issues relating to the rigor of both the MS and the modeling studies, and think that the data is well presented. My concern, rather, is that there is a small, but significant, fraction of crosslinks that fall beyond expected distances regardless of methodology (x-ray, cryo, or, in this case, AlphaFold2 predicted structures). This is not a critique of MS nor of AF2, just that there are crosslinks that do not fit into known structures. As such, I feel more comfortable in stating that the concordance between the crosslinking studies and the predictive structures show the capability of AF2 in modeling biologically relevant structures. At what point "concordance" becomes "validation" is something that I, and I think the field, struggles with. I think the revised version addresses many of these concerns. As noted in my initial review "A paper reporting the structure using predictive methods and XLMS together to predict the structure of endogenous ciliary proteins of unknown overall structure would be more favorably reviewed". I consider the revised version to be such a paper.

Given the above comments, I would suggest a single additional change in the manuscript. In the second sentence of the Conclusion I suggest using "concordance" rather than "general correctness", as the latter term is ill-defined and not used previously in the manuscript.

Responses to referees

McCafferty *et al.*, Does AlphaFold2 model proteins' intracellular conformations? An experimental test using cross-linking mass spectrometry of endogenous ciliary proteins

Our responses follow in-line in blue text

Editor's comments:

Your manuscript entitled "Does AlphaFold2 model proteins' intracellular conformations? An experimental test using cross-linking mass spectrometry of endogenous ciliary proteins" has now been seen by 3 referees, whose comments are appended below. You will see from their comments copied below that while they find your work of potential interest, they have raised concerns that must be addressed. In light of these comments, we cannot accept the manuscript for publication in its current form, but would be interested in considering a revised version that addresses these concerns.

Please address all comments from the reviewers. In response to Reviewer 3, please change the language of the paper from using XLMS for experimental validation to using it for prediction and refinement of structures.

We also request validation of XLMS structures when the AlphaFold2 model has crosslink violations. Specifically:

- Validation of the XLMS-refined model in Figure 4 whether experimentally or computationally, such as through docking or MD simulations.
- Validation of the XLMS-refined model in Figure S2: if the violations are fixed, are the other crosslinks still satisfied?

We thank the editor for handling the above referenced manuscript, and their clear guidance and suggestions for the response. We have addressed each of the concerns of Reviewers 1 and 2 in-line below. With respect to Reviewer 3, as we have in fact performed experimental validation, not prediction, we opted instead to more extensively detail the (considerable) literature support for this use case. As requested, we also added additional support for the model of Figure 4. We now include a brief methods section for the modeling and a new supplementary figure (Figure S4), which plots the distributions of cross-linker lengths before and after integrative modeling as well as provides supporting statistics from the integrative modeling procedure that indicate model convergence and overall confidence. Regarding Figure S2, note that we did not perform XLMS-refinement; this figure simply reports the concordance between the experimental XL/MS data and the AlphaFold-predicted model, where it is evident that the only disagreement is in the relative positioning of well-folded (and concordant) domains. We have clarified the legend of Figure S2 accordingly.

For the benefit of the editor we reproduce the new Figure S4 here:

Figure S4. Sampling exhaustiveness and improvement of the BBC118 model using the DSSO cross-links. **a)** Integrative modeling of BBC118 produced a total of ~30,000 high-scoring models. 29,000 models make up the largest cluster with a cluster precision of 3.369 \square . **b)** Randomly splitting the models into 2 samples and assessing the samples to determine if they come from the same parent distribution confirms that the models' score distributions are similar (small K-S test D), indicating that the models had converged. **c)** The sampling precision is 4.760 \square as defined and explained in (55). **d)** The distribution of cross-link distances before and after modeling indicate that the modeled conformation of BBC118 could satisfy nearly all of the cross-links.

Reviewers' comments:

Reviewer #1 (Remarks to the Author):

The submitted work from McCafferty et. al. is a nice validation of predicted AlphaFold2 model proteins using cross-linking mass spectrometry. I have few comments which might of help to increase clarity.

A. Page 4, Last paragraph - These results present how well AF2 models work and could be presented in a table.

We thank the reviewer for this comment and have now incorporated the data from the final paragraph into Table S2.

B. Figure S2 - Colors in fig K could have more contrast for clarity.

We have now enhanced the contrast of the colors in Figure S2.

Reviewer #2 (Remarks to the Author):

Authors Caitlyn L. McCafferty et al. in this manuscript have investigated whether AlphaFold2 structure predictions match native protein conformations using the cross-linking mass spectrometry of endogenous ciliary proteins for experimental validation. Authors have performed chemical cross-linking of ciliary proteins in situ within intact *Tetrahymena thermophila* and native enriched extracts using the membrane-permeable, mass-spectrometry cleavable chemical cross-linker disuccinimidyl sulfoxide (DSSO). Authors have found 1,225 intramolecular cross-links within the 100 best-sampled proteins to benchmark the distance restraints obeyed by proteins in their native assemblies.

Using this protein set, authors have achieved good concordance of about 86.2% between our cross-links and AF2 structure predictions, with some disagreements or violations between domains of multi-domain proteins and dynamic proteins of conformational change. Authors have shown that 13.8% of cross-link violations occurred in Proteins with the average predicted local distance difference test (pLDDT) confidence score below 70. The remainder of the violations also occurred in reasonably confident protein structures (pLDDT over 70). Here, authors have come up with a unique calibration between predicted aligned error (PAE) scores produced by AF2 and the experimentally obtained chemical-cross link violations, by binning the cross-linked amino acids across all 100 proteins according to their PAE scores which established a linear relationship between the PAE and the cross-links, with larger PAE values correspond to a larger proportion of cross-link violations. Authors have showed AF2's high confidence is appropriate in this regime with no crosslink violations in the PAE range of 0 to 3.5.

Overall, the manuscript, and the detailed systematic analysis shows promising results for investigating more case studies and challenges posed by AF2 prediction and interpret the relative quality of specific regions of AF2 structures in native conformations.

We thank the referee for this excellent distillation of the paper.

Minor comments

- The title "Does AlphaFold2 model proteins' intracellular conformations?" takes me to the question raised from the following statement in page 4, "Impressively, 43% of AF2 predicted

protein structures show no disagreements with the in situ cross-links, and a large majority (87%) showed three or fewer cross-link violations, demonstrating AF2 predicts biologically relevant protein conformations.

Here, it is not clear whether agreement between in situ-crosslinks and AF2 prediction is the governing factor for structure that's biologically relevant protein conformation? Or the AF2 prediction as it incorporates information from evolutionary coupling and amino acid conservation which should, in principle, capture structures most relevant to the predominant cellular roles of these proteins.

We thank the reviewer for raising this question. We used our *in situ* XL/MS data as a direct experimental validation of amino acid positions because it can capture protein arrangements as they exist in the cell (including dynamics). We therefore consider the agreement between AF2 prediction and our XL/MS data as a validation of AF2's ability to produce biologically relevant conformations of proteins. As to how AF2 arrives at these structures, it is indeed likely that at least part of AF2's ability to capture these biological conformations may derive from the evolutionary coupling data used in building the models. We now emphasize this point on p. 3 and again on p. 4.

- Also, the authors have not discussed about any other chemical cross-linker molecule other than DSSO. Is there any other mass spectrometry cleavable linker that can be used for this study? If so, does it alter the cross-linking coverage or violations

Other cross-linkers are indeed available that might be used (although their application would be beyond the scope of our goals here, which was to assess AlphaFold-predicted structures using independently obtained experimental protein structure data for a large number of proteins). To address this comment, as well as comments by Reviewer 3 below, we now add a section to the paper on p. 3 with discussion and literature review of other cross-linkers and their applications to structure validation.

- Is there a concentration dependence for cross-linker in case higher oligomers or assembly protein such as outer dynein arms (ODA) proteins.

This is an interesting question for which we have no simple answer. There is definitely a strong concentration dependence to which sites are cross-linked, and having many replicate copies of the same interface certainly increases the chances of observing the cross-links. However, it is not clear that higher order assemblies should be any more likely to be cross-linked than the same number of e.g. dimers in solution. It is more likely the concentration of the interfaces that matters for these assays, not the overall oligomeric state of the assemblies, unless the cross-linkers themselves should e.g. promote oligomerization. We have seen no evidence of this. We now briefly note the effect of concentration on p. 4.

Reviewer #3 (Remarks to the Author):

The submitted manuscript describes efforts to validate AlphaFold2's prediction of endogenous ciliary proteins using chemical crosslinking coupled with MS methodology, as determined crosslink distances will be used to test the predicted models. The paper is well written, but I found the overall hypothesis to be fatally flawed. The authors state "Global benchmarking and independent validation of such predicted structures will be necessary to inform reliable and nuanced interpretations of these structures" and posits crosslinking to supply such benchmarking. The distances found in XLMS studies often does not correspond with known structures (for reviews, see Graziadei and Rappsilber, Structure 2021 and Mintseris and Gygi, PNAS 2019). While XLMS is a good complement, it is not yet appropriate for benchmarking purposes and cannot be used for experiment validation. A paper reporting the structure using predictive methods and XLMS together to predict the structure of endogenous ciliary proteins of unknown overall structure would be more favorably reviewed, but it appears that the focus of this paper greatly overstates the validation aspects. As such, I find the paper as written to be unacceptable for publication.

The referee makes quite a strong statement here ("I found the overall hypothesis to be fatally flawed"), and we wish to be quite sure that we understand correctly what is being objected to.

If the referee means that they believe that it is not possible to use XL/MS to measure *specific* distances between alpha carbons (*i.e.* to state that two residues must be precisely 30 Angstroms apart based on the cross-links), then we agree completely, and please note this is not what we are doing. Just in case this is what the referee was indeed referring to, we have now clarified the language on p. 4 to help other readers to avoid this interpretation.

Rather, we treat each crosslink as a distance *restraint*, *i.e.* a specific threshold distance that should not generally be exceeded. Chemical crosslinks correspond to actual physical entities with known atomic structures of fixed maximal lengths connecting pairs of amino acid residues. Leaving aside for a moment the issues of assignment errors, the direct experimental observation of a crosslink implies that those two amino acids *must* have resided within the threshold distance for sufficient time and in a sufficient proportion of proteins in the population to be *robustly* chemically coupled and subsequently detected by mass spectrometry, implying that e.g. more than 100 million copies of the protein(s) were observed to adopt the necessary conformational state to be cross-linked, based on a typical (~fmol) detector sensitivity. As this is a very high bar in general to clear, the method shows strong abundance biases and preferences for long-lived, stable 3D protein conformations.

However, the referee's subsequent statement that "XLMS ... is not yet appropriate for benchmarking purposes and cannot be used for experiment validation" is at odds with a decade of research and a large and growing number of successful use cases across the field.

This is such a fundamental point that it is worth elaborating on.

Importantly, crosslinks are not predictions, but *independent experimental observations*. As with any such set of physical measurements, there will be some proportion of measurement and attribution errors, and protein dynamics will affect proximity as well. However, these effects have been well-studied in the literature, and crosslinks are widely appreciated across the field as important experimental validations of protein models. For example, Brodie *et al.*, in *Science Advances* 3(7):e1700479 (2017), explicitly state:

"Long-distance Lys-Lys cross-linking using amine-reactive reagents with spacer lengths of >10 Å ... can be used to validate [predicted] protein structures".

Similarly, Yu and Huang note in *Anal. Chem.* 90(1):144–165 (2018):

"cross-links can serve as direct evidence for pairwise protein interactions without further biochemical confirmation, as often required for conventional AP-MS experiments" and "cross-links can be utilized as distance constraints for various applications ... [including] structure validation and integrative modeling."

In fact, the two reviews cited by the referee explicitly support this use case:

(1) Graziadei and Rappsilber, *Structure* 2022, state:

"The capacity of crosslinking MS to provide distance restraints in complex samples, or even in cells, means that it has proven to be an ideal complementary technique for validating EM or crystallographic models in their native context (Casañal *et al.*, 2019; O'Reilly and Rappsilber, 2018; Schmidt and Urlaub, 2017)."

As this is quite a recent review (Jan 2022), the authors even note the potential for our specific application:

"Recently, there have been breakthroughs in protein structure prediction by the deep learning algorithms AlphaFold2 (Jumper *et al.*, 2021; Tunyasuvunakool *et al.*, 2021) and RosettaFold (Baek *et al.*, 2021). Crosslinking-MS may be used to understand the arrangement of multidomain proteins and validate or select between proteins with different arrangements of low-confidence regions."

They conclude that crosslinks

"can be used to validate models, map binding interfaces, or derive new models by docking and integrative modeling."

(2) Mintseris and Gygi, *PNAS* 2019, perform their own comparison of structure and cross-links and find excellent concordance, including for the BS3 crosslinker, the closest match to our DSSO crosslinker, noting that:

“cross-linking data could be used to validate candidate cryo-EM models.”

The authors compute a structure-based FDR on their crosslinks (see Fig. S4), from which they

“estimate a value of 3.2% structure-based FDR for the combined dataset with a 95% confidence interval (2.0 and 4.5%)”.

That is to say, they observe a 97% concordance with independently-derived experimental structures.

As to the issue of protein dynamics, we refer the referee in particular to the study of Merkle *et al.*, *Protein Science* 23:747—759 (2014), whose study of both experimental and modeled structures lead them to conclude that even with the consideration of protein dynamics, that:

“~84% of the observed intramolecular crosslinks and ~86% of the observed intermolecular crosslinks have C α distances smaller than the theoretical (i.e., both linker and side chains fully extended) C α –C α distance of 24 Å, suggesting that the common 24-Å criterion is in fairly close agreement with the data” and that “Our estimated maximum crosslinking distance values (24–30 Å between C α atoms) agree well with the experimental distance distribution. For C α –C α distances, 89.3% of the observed crosslinks fall below our recommended maximum threshold of ~30 Å. Our recommended C α threshold value is similar to the values applied by Aebersold and coworkers (30 Å) and only slightly longer than the value applied by Sinz and coworkers (27.4 Å). Rappsilber and coworkers have used 27.4 Å, but more recently they suggest a range (25–29 Å). Thus, the Dymeomics-based maximum C α recommendation is in line with the empirically determined threshold values already in common use.”

Importantly, all of these authors describe a significant concordance across the field for the use of crosslinks as experimental distance restraints and for independent structural validation, as well as for the specific 30 Angstrom threshold value we employ.

We have therefore taken the referee’s comments to indicate that we should do a better job at communicating this well-established utility in the revised paper and at citing other examples using XL/MS to independently validate models derived by other approaches. To address these points, we now add the above references and include a condensed discussion of these points on p. 3.

REVIEWERS' COMMENTS:

Reviewer #3 (Remarks to the Author):

I appreciate the changes made by the authors, as well as the care they have taken to

respond to my initial review. I do consider that we have a common interest in utilizing MS crosslinking restraints for integrative structural biology. My concerns were pointedly not to question the validity of measuring the outer bound of distance restraints in native crosslinking studies, and we cited common reviews showing the validity of crosslinking MS and its ability to complement other high resolution studies. I raised no issues relating to the rigor of both the MS and the modeling studies, and think that the data is well presented. My concern, rather, is that there is a small, but significant, fraction of crosslinks that fall beyond expected distances regardless of methodology (x-ray, cryo, or, in this case, AlphaFold2 predicted structures). This is not a critique of MS nor of AF2, just that there are crosslinks that do not fit into known structures. As such, I feel more comfortable in stating that the concordance between the crosslinking studies and the predictive structures show the capability of AF2 in modeling biologically relevant structures. At what point “concordance” becomes “validation” is something that I, and I think the field, struggles with. I think the revised version addresses many of these concerns. As noted in my initial review “A paper reporting the structure using predictive methods and XLMS together to predict the structure of endogenous ciliary proteins of unknown overall structure would be more favorably reviewed”. I consider the revised version to be such a paper.

Given the above comments, I would suggest a single additional change in the manuscript. In the second sentence of the Conclusion I suggest using “concordance” rather than “general correctness”, as the latter term is ill-defined and not used previously in the manuscript.

We thank the reviewer for this comment and have made the suggested change in the Conclusion on pg. 6.